# Semantic Codebook Learning for Dynamic Recommendation Models

## ABSTRACT

Dynamic Sequential Recommendation (DSR) systems, which adapt to users' changing preferences by adjusting their parameters in real time, have emerged as a significant advancement over traditional static models. Despite their advantages, DSR systems face challenges including large item parameter spaces and heterogeneous user-item interactions, which can destabilize the recommendation process. To address these issues, we introduce the Semantic Codebook Learning for Dynamic Recommendation Models (SOLID) framework. SOLID compresses the parameter generation model's search space and utilizes homogeneity within the recommendation system more effectively. This is achieved by transforming item sequences into semantic sequences and employing a dual parameter model, which combines semantic and item-based cues to tailor recommendation parameters. Our innovative approach also includes the creation of a semantic codebook, which stores disentangled item representations to ensure stability and accuracy in parameter generation. Through extensive testing, SOLID has shown to surpass traditional DSR systems, providing more precise, stable, and dynamically adaptable recommendations. [1].

## CCS CONCEPTS

• **Computing methodologies**; • **Information systems** → **Personalization**;

## KEYWORDS

Disentangled, Recommendation System, Multimodal

## 1 INTRODUCTION

Today, recommendation systems based on deep learning have rapidly evolved from traditional Collaborative Filtering (CF) methods to sequence recommendations that suggest items based on a user's recent behavior sequence. This evolution has led to the emergence of numerous well-known sequential recommendation models, such as DIN [29], GRU4Rec [9], SASRec [11], and BERT4Rec [17]. However, the behavior logic of most users is not universally applicable, and as interests can change, it necessitates that sequence recommendation models be able to adjust their parameters in real-time according to the user's current interest preferences. Consequently,

---

[1]Our source code can be referred to https://anonymous.4open.science/r/SOLID-0324

dynamic sequential recommendation models (DSR) like DUET [13] and APG [26] have been developed.

The DSR paradigm consists of two parts: (1) The primary model. This model has a structure similar to conventional sequential recommendation models like SASRec, but it is divided into a static layer and a dynamic layer. The parameters of the static layer remain unchanged after pre-training, whereas the parameters of the dynamic layer change with the user's behavior. (2) The parameter generation model. This is mainly used to sparse user behavior and generate the parameters for the dynamic layer of the primary model based on this behavior. The DSR paradigm enables traditional static sequential recommendation models to quickly adjust their parameters according to the potential shift of interests and intentions reflected in user behaviors, thus dynamically obtaining more interest-aligned models in real time.

Despite the promising potential of Dynamic Sequential Recommendation (DSR) systems, they face significant challenges, primarily stemming from the item-to-parameter modeling scheme: (1) A large number of items result in a vast search space for the parameter generation model. Slight variations in user behavior sequences, such as "shirt, tie, suit" versus "tie, shirt, suit," which suggest similar preferences, can unpredictably alter the item-to-parameter modeling, introducing complexity and potential instability. (2) The interaction between users and items is generally sparse and potentially noisy (*e.g.*, the notorious implicit feedback issue), leading to heterogeneous behavior sequences that complicate the learning of accurate item representations. This results in inaccurate item representation learning, weakening the precision of model parameter customization based on item sequence features, and further exacerbating the inaccuracy of generated parameters.

To address these issues, we propose the **S**emantic C**o**debook **L**earning for Dynam**i**c Recommen**d**ation Models (SOLID). The core objective of SOLID is to compress the search space of the parameter generation model, promoting homogeneity signals utilization within the recommendation system. We construct a semantic codebook that better utilizes these homogeneity signals. In this codebook, item representations are disentangled into semantics that are learned to be absorbed in the codebook elements, such that the homogeneity between items in the disentangled latent space can be established. The user-item interactions are transformed into density-enriched user-semantic interactions in the latent space. The enriched density reduces the heterogeneity and complexity of user behavior space modeling in the parameter generator. Moreover, SOLID shifts from a traditional item sequence-based parameter model to a dual approach (item sequence + semantic sequence) → model parameter, effectively merging both uniform and diverse information in a structured manner. Uniform information derived from the semantic-to-parameter part is utilized to develop parameters that generalize across certain user behaviors, while diverse information allows for the crafting of specific parameters tailored

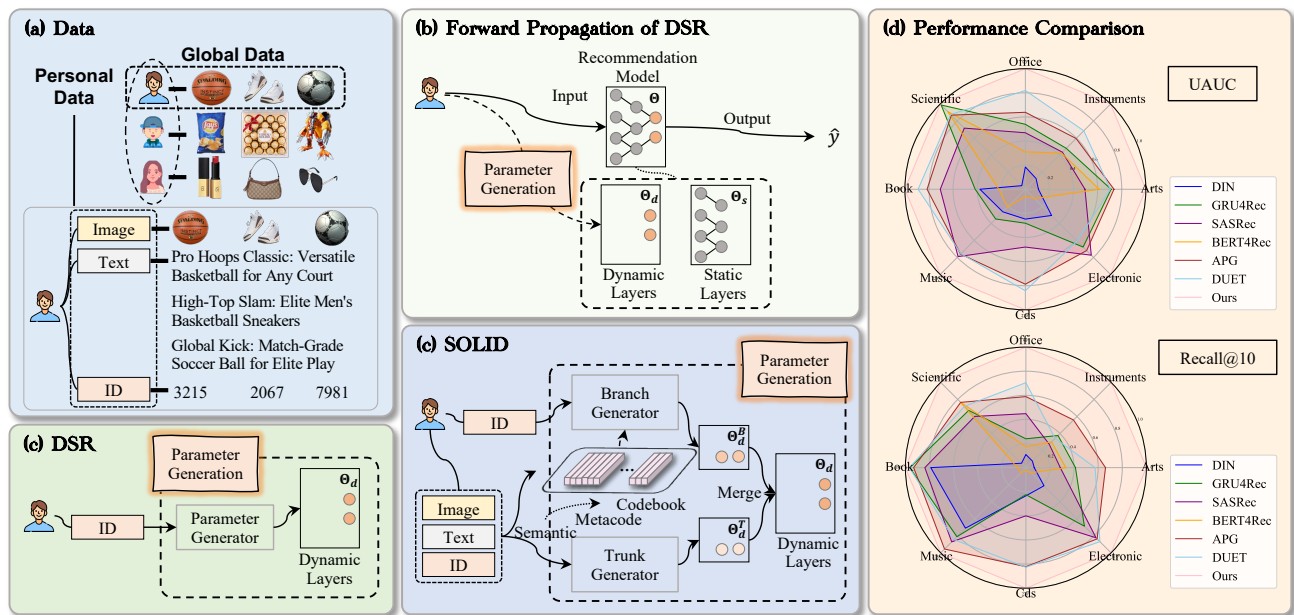

Figure 1: (a) describes multimodal user behavior data that includes images, text, and IDs. (b) describes the forward propagation of DSR, which is divided into two pathways: the first pathway processes user behavior data composed of IDs through a parameter generator to produce the parameters for the dynamic layers of the primary model. The second pathway processes the same ID-based user behavior data through the primary model's static layer, then through the dynamic layer, resulting in the prediction output. (c) and (d) compare the parameter generation patterns of existing DSR and SOLID. (e) compares the performance of our method and SR models and DSR Models on four multimodal recommendation datasets and four additional recommendation datasets. The results show that our method significantly enhances performance on extensive datasets.

to individual behavioral nuances. Crucially, by aligning the dimensions of the codebook with those of the semantic encoder, we transform the semantic encoder into a meta-code that serves as an initial state for the codebook, further easing the modeling of parameter generation.

Specifically, to reduce the search space of the parameter generation model through the semantic codebook, SOLID involves three main modules. Initially, SOLID employs a pretrained model to extract semantic components from item, image, and text features. This disentanglement transitions the focus from item sequences to semantic sequences, shifting the modeling approach from item-based to semantics-based parameter generation. This design results in trunk parameters that generalize behaviors from the entire user base to specific groups, and branch parameters that cater to individual user behaviors, both derived from semantic and item sequences respectively. Parameters derived from items are tightly controlled (e.g., ±0.01) before their integration into the dynamic layer of the primary model, ensuring a responsive and adaptive system based on real-time user activity. Despite this, branch parameters still adhere to an item-centric approach, necessitating the use of a Semantic Codebook (SC) to maintain personalization and stability in representation. This codebook stores semantic vectors of behavior, progressively aligned with the nearest matches during learning. The weights of the semantic encoder are used to initialize the SC, easing the semantic codebook learning. As demonstrated, SOLID is designed to pursue the precision, stability, and clarity of model

parameter generation, trying to promote the dynamic recommendation model's response to sparse, heterogeneous, and potentially noisy user behaviors.

Our contributions can thus be summarized as:

- We identified the limitations of the existing Dynamic Sequential Recommendation (DSR) paradigm and designed the SOLID framework with a semantic codebook to address these deficiencies.
- We first learned to disentangle the parameter generation model, vertically splitting the item-to-parameter scheme into semantic-to-parameter as the trunk and item-to-parameter as the branch. This approach ensures that the generated model parameters contain both homogeneous and heterogeneous information.
- To enhance the semantic codebook learning, we transformed the semantic encoder used for trunk parameter generation into a semantic metacode, which is used to initialize the semantic codebook.
- We conducted extensive experiments across multiple datasets, where varied analysis demonstrates the rationality and efficacy of SOLID.

## 2 RELATED WORK

### 2.1 Sequential Recommendation

Sequential recommendation systems analyze users' historical interactions to predict future preferences, playing a crucial role in various applications like e-commerce and short video platforms.

Early approaches, such as the FPMC model [15], leveraged non-deep learning techniques and Markov decision processes to understand user behavior sequences. However, to enhance model capabilities, recent advancements [4, 9, 11, 17, 24, 29] have shifted towards deep learning-based sequential recommendation systems. For instance, GRU4Rec [9] employs Gated Recurrent Units (GRUs) to effectively model sequential behavior, demonstrating impressive results. Additionally, models like DIN [29] and SASRec [11] incorporate attention mechanisms and transformers, respectively, to achieve rapid and efficient performance enhancements. BERT4Rec further applies the principles of BERT for superior outcomes. These four models—GRU4Rec, DIN, SASRec, and BERT4Rec—have significantly influenced both academic research and industry practices with their innovative approaches to sequential recommendation. However, these SR Models struggle to achieve optimal performance across every data distribution when dealing with users' real-time changing behaviors and interest preferences. Therefore, we have designed a more robust DSR Model that retains the advantages of existing DSR Models while addressing their shortcomings.

## 2.2 Disentangled Representation Learning

Disentangled Representation Learning: The goal of disentangled representation learning is to parse the data into distinct, interpretable components by identifying different underlying latent factors [2, 3]. The Variational Autoencoder (VAE) [6], a cornerstone in this field, employs variational inference alongside an encoder-decoder structure to differentiate between various latent factors. The $\beta-$VAE model [10] further refines this by adjusting the balance between the model's ability to disentangle and its capacity to represent information. This approach has found notable applications in areas like recommendation systems, where it addresses the diverse purchasing preferences of users. By incorporating multi-interest methods [12, 14] along with disentangled representation learning, several studies [20–23, 28] have demonstrated significant advancements in recommendation tasks. We draw on the idea of disentangling and apply it to dynamic model parameter generation to reduce the parameter search space and leverage the homogeneous information of user behavior.

## 2.3 Dynamic Neural Network

Research in dynamic neural networks unfolds across two primary branches: HyperNetworks and Dynamic Filter Networks. HyperNetworks, initially introduced by Ha et al. [8], represent a novel approach wherein one neural network dynamically generates the parameters for another, offering a significant reduction in the number of parameters required for training, thus achieving model compression. This foundational concept has sparked extensive exploration into various applications and enhancements of HyperNetworks. For instance, Oscar et al. [5] delved into parameter initialization techniques specifically for HyperNetworks. The versatility of HyperNetworks has been demonstrated across a wide array of tasks, including continual learning [18], graph analysis [27], meta-learning [25], and federated learning [16]. Recent advancements have particularly focused on leveraging HyperNetworks to generate unique network parameters based on differing data inputs, with HyperStyle [1] and HyperInverter [7] showcasing the potential for enhancing image

reconstruction quality by producing distinct decoder parameters for various images. APG [26] and DUET [13] are the most recently and the SOTA dynamic recommendation models. However, existing DSR models are affected by the heterogeneity of user behavior, the sparsity of user-item interactions, etc., leading to drawbacks such as an overly large parameter search space and inaccurate parameter generation. Our method effectively addresses these shortcomings of the DSR models.

## 3 METHODOLOGY

### 3.1 Notations and Problem Formulation

First, we introduce the notation in sequential recommendations.

*3.1.1 Data.* We use $\mathcal{X}_{\text{ori}} = \{u, v, s_v\}$ to represent a piece of data, $\mathcal{X}_{\text{dec}} = \{u, c, s_c\}$ to represent a piece of disentangled data, $\mathcal{X}_{\text{mm}} = \{i, t\}$ to represent multimodal information, and $\mathcal{Y} = \{y\}$ to represent the label indicating whether the user will interact with the item. In brief, $\mathcal{X} = \mathcal{X}_{\text{ori}} \cup \mathcal{X}_{\text{dec}} \cup \mathcal{X}_{\text{mm}} = \{u, v, s_v, c, s_c, i, t\}$, where $u, v, c, s_v, s_c, i, t$ represent user ID, item ID, category ID, user's click sequence consists of item ID, user's click sequence consists of category ID, the image of the item, and the title of the item respectively. We represent the dataset as $\mathcal{D}$, where $\mathcal{D} = \{X, Y\}$. More specifically, we use $\mathcal{D}_{\text{Train}}$ to represent the training set and $\mathcal{D}_{\text{Test}}$ to represent the test set. Roughly speaking, let $\mathcal{L}$ be the loss obtained from training on dataset $\mathcal{D}_{\text{Train}}$. For simplicity, we simplify the symbol $\mathcal{D}_{\text{Train}}$ to $\mathcal{D}$. Then, the model parameters $W$ can be obtained through the optimization function $\arg \min \mathcal{L}$. The sequence length inputted into the model is set to $L_s$, so the lengths of both $s_v$ and $s_c$ in a sample are $L_s$.

*3.1.2 Model.* The recommendation model is represented by $\mathcal{M}$ and the parameters of the $\mathcal{M}$ is $\Theta$, where $\Theta = \Theta_s, \Theta_d$. The model $\mathcal{M}_v$ is utilized to generate the $\Theta_d$ according to the item id sequence $s_v$, $\mathcal{M}_c$ is utilized to generate the $\Theta_d$ according to the category id sequence $s_c$, $\mathcal{M}(\cdot)$ and $\mathcal{M}_v(\cdot)$ represent the forward propagation processes of two models, where $\cdot$ denotes the input.

*3.1.3 Feature.* We use $\mathbf{E_v}$ and $\mathbf{E_c}$ to represent the item feature set and semantic feature set extracted from $s_v$ and $s_c$ respectively. Specifically, $\mathbf{E_v} = \{e_v^1, e_v^2, ..., e_v^{L_s}\}$, $\mathbf{E_c} = \{e_c^1, e_c^2, ..., e_c^{L_s}\}$. $\mathbf{e_v}$ and $\mathbf{e_c}$ are the sequence features obtained through sequence feature extraction models such as Transformer or GRU, via $\mathbf{E_v}$ and $\mathbf{E_c}$, respectively. The length of an item representation or a semantic representation is set to $L_r$.

*3.1.4 Formula.* Sequential Recommendation Models (SR), Dynamic Sequential Recommendation Models (DSR), and Disentangled Multimodal Dynamic Sequential Recommendation Models (SOLID) can be formalized as follows:

$$\textbf{SR}: \underbrace{\mathcal{M}(\mathcal{X}_{\text{ori}}; \Theta)}_{\text{Recommendation Procedure}} \underset{\text{Output}}{\overset{\text{Gradients}}{\Longleftarrow\!\!\!=\!\!\!=\!\!\!\Longrightarrow}} \underbrace{(\hat{\mathcal{Y}} \Longleftrightarrow \mathcal{Y})}_{\text{Loss Calculation}}. \quad (1)$$

$$\textbf{DSR}: \underbrace{\mathcal{M}(\mathcal{X}_{\text{ori}}; \Theta_s, \Theta_d = \mathcal{M}_v(\mathcal{X}_{\text{ori}}))}_{\text{Recommendation Procedure}} \underset{\text{Output}}{\overset{\text{Gradients}}{\Longleftarrow\!\!\!=\!\!\!=\!\!\!\Longrightarrow}} \underbrace{(\hat{\mathcal{Y}} \Longleftrightarrow \mathcal{Y})}_{\text{Loss Calculation}}. \quad (2)$$

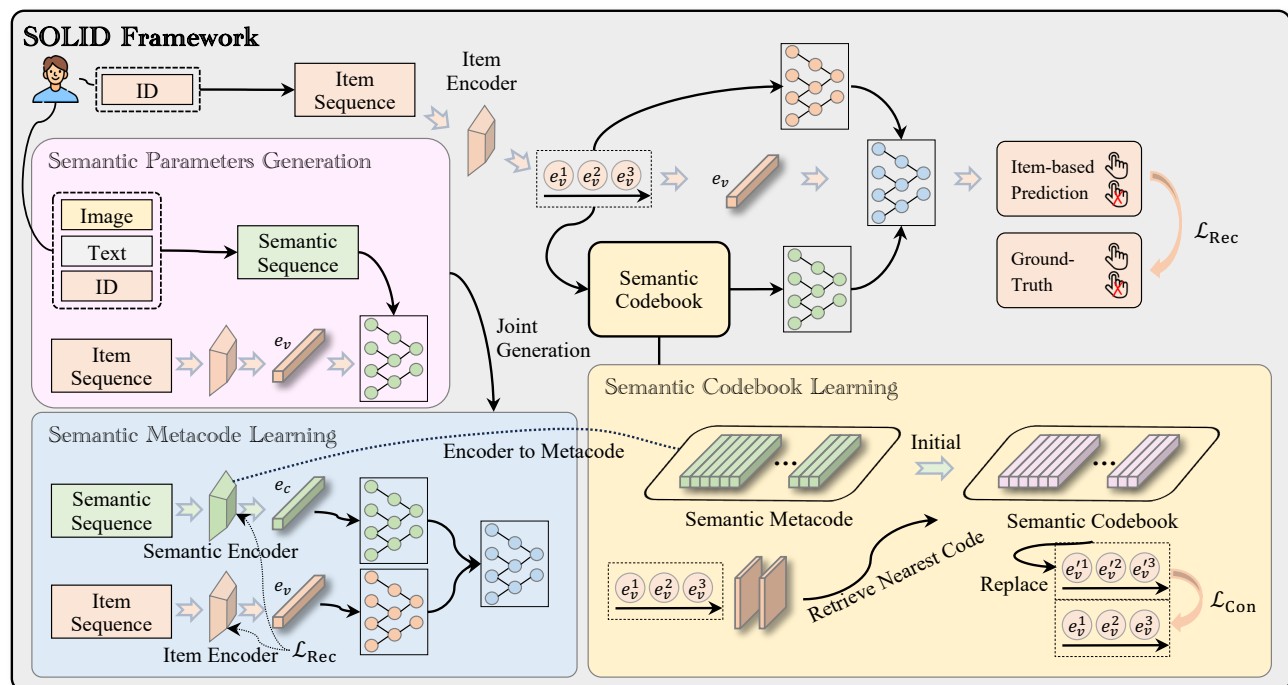

**Figure 2: The framework of the SOLID, which consists of three main modules: Semantic Parameter Generation (SPG), Semantic Metacode Learning (SML), and Semantic Codebook Learning (SCL). SPG first converts item representations into semantics and constructs a semantic sequence to generate parameters in a structured manner. Subsequently, SML generates model parameters based on both the item sequence and the semantic sequence, and it jointly trains the model, accommodating both homogeneous and heterogeneous information. More importantly, the semantic encoder it learns can be transformed into metacode, which then provides a good initial value for the codebook. Finally, SCL learns a semantic codebook to improve the process of the parameter generation. Among them, $\mathcal{L}_{\mathbf{Rec}} = l_{\mathbf{CE}}(y, \hat{y})$, $\mathcal{L}_{\mathbf{Con}} = l_{\mathbf{MSE}}(\mathbf{E}_v, \mathbf{E}'_v)$.**

$$\mathbf{SOLID}: \begin{cases} \mathcal{X}_{\text{ori}}, \mathcal{X}_{\text{mm}} \mapsto c = f(v, i, t) \mapsto \mathcal{X}_{\text{dec}}, \\ \Theta_d = \mathcal{M}_v(\mathcal{X}_{\text{ori}}) \oplus \mathcal{M}_c(\mathcal{X}_{\text{dec}}), \\ \underbrace{\mathcal{M}(\mathcal{X}_{\text{ori}}; \Theta_s, \Theta_d)}_{\text{Recommendation Procedure}} \xleftarrow[\text{Output}]{\text{Gradients}} \underbrace{(\hat{\mathcal{Y}} \Longleftrightarrow \mathcal{Y})}_{\text{Loss Calculation}}. \end{cases}$$
$$(3)$$

In the aforementioned formula, $a \rightarrow b$ indicates that the direction of information transfer is from $a$ to $b$, with the text next to it representing the content of the transfer. $a \mapsto b$ signifies that $b$ is derived from $a$.

## 3.2 Preliminary

### 3.2.1 Sequential Recommendation Models.
Here we first retrospect the paradigm of sequential recommendation.

In the training stage, the loss can be calculated to optimize the sequential recommendation models as follows,

$$\min_{\Theta} \mathcal{L}_{\text{SR}} = \sum_{u,v,s_v,y \in \mathcal{D}} l_{\text{CE}}(y, \hat{y} = \mathcal{M}(u, v, s_v; \Theta)). \quad (4)$$

The loss function can set to CE (Cross Entropy) loss and MSE (Mean Squared Error) loss, etc. However, since sequential recommendation often focuses more on CTR (Click-Through Rate) prediction tasks, and this paper is also focused on CTR prediction, the recommendation loss in this paper is CE loss and represented by $l_{\text{CE}}$.

### 3.2.2 Dynamic Sequential Recommendation Models.
DSR generate model parameters based on users' real-time user behaviors. Then the updated model is used for current recommendations. The layers whose parameters are updated are referred to as "dynamic layers", while the layers whose parameters remain unchanged are referred to as "static layers".

DSR treat the parameters of one of the dynamic layers as a matrix $K \in \mathbb{R}^{N_{in} \times N_{out}}$, where $N_{in}$ and $N_{out}$ represent the number of input neurons and output neurons of a fully connected layer (FCL), respectively. DSR utilize a encoder $E_v$ to extract the sequence feature $\boldsymbol{e}_v$ from the user's behavior sequence $s_v$ to generate the parameters of the model's dynamic layers.

$$\theta_d = \mathcal{M}_v(E_v(s_v)), \quad (5)$$

After parameter generation, the parameters of the model will be reshaped into the shape of $K$.

During training, all layers of the $\mathcal{M}_v$ are optimized together with the static layers of the $\mathcal{M}$. The loss function $\mathcal{L}_{\text{DSR}}$ is defined as follows:

$$\min_{\Theta_s, \Theta_v} \mathcal{L}_{\text{DSR}} = \sum_{u,v,s_v,y \in \mathcal{D}} l_{\text{CE}}(y, \hat{y} = \mathcal{M}(u, v, s_v; \Theta_s, \Theta_d)). \quad (6)$$

Although the Item-based Dynamic Recommendation Model can obtain personalized model parameters based on users' real-time behavior and achieve superior performance, it also faces multiple

challenges. 1) The user-item interaction is extremely sparse, leading to inaccurate item representation learning, making the model parameters customized based on item-based features inaccurate. 2) The personalized model parameters obtained by this strategy are highly mixed. 3) The generated parameters are not subject to any constraints, which poses challenges to the stability of the generated model. So we design the novel methods to address the challenges mentioned above.

### 3.3 SOLID Framework

Our proposed Semantic Codebook Learning for Dynamic Recommendation Models (SOLID), which is shown in the Figure 2.

*3.3.1 Semantic Parameter Generation.* Transforming the Item-based Dynamic Recommendation Model into a Semantic-based Dynamic Recommendation Model is an important step in disentangling personalized model parameters.

`Item to Semantic.` First, items need to be transformed into semantics. For data without category labels, clustering can be directly applied to obtain semantics, i.e.,

$$\text{Cluster}(\{e_i\}_{i=1}^{\mathcal{N}}) \mapsto \{c_i\}_{i=1}^{\mathcal{N}}, c_i \in \{1, 2, ..., k\}. \quad (7)$$

For data with category labels, since the same item often belongs to multiple categories, we need to select a primary category as semantic it. First, we can define the centroid $m_c$ of each category $c$, which is the average of embeddings $e$ for all items belonging to category $c$. Assuming $n_c$ is the number of items belonging to category $c$, the centroid $m_c$ for category $c$ can be represented as:

$$m_c = \frac{1}{n_c} \sum_{v \in c} (e_v \text{ or } e_i \text{ or } e_t), \quad (8)$$

where $e_v, e_i, e_t$ are the representation of item ID $v$, item image $i$, item title $t$, respectively. Next, we compute its distance to each category center $m_c$. Assuming we use the Euclidean distance, it can be represented as,

$$d(v, c) = \|(e_v \text{ or } e_i \text{ or } e_t) - m_c\|, \quad (9)$$

where $\|\cdot\|$ denotes the norm of the vector, typically the Euclidean norm. Finally, we select the closest category as the semantic for item $v$. That is, the semantic $c_p$ for item $v$ can be represented as:

$$c_p = \arg\min_c d((v \text{ or } i \text{ or } t), c). \quad (10)$$

This way, we can use formulas to represent the process of computing the primary category for an item.

`Semantic-based Parameter Generation.` After converting items into semantics, a semantic-to-parameter model can be trained. The training process is similar to that of the item-to-parameter model. The only differences are that the input for the item-to-parameter model is an item sequence, whereas for the semantic-to-parameter model, it is a semantic sequence; similarly, the outputs are the target item and target semantic, respectively.

$$\begin{cases} \min_{\Theta_s, \Theta_c} \mathcal{L}_{\text{DSR}} = \sum_{u,v,s_c,y \in \mathcal{D}} l_{\text{CE}}(y, \hat{y}), \\ \hat{y} = \mathcal{M}(u, v, s_c; \Theta_s, \Theta_d), \\ \Theta_d = \mathcal{M}_c(E_c(s_c)). \end{cases} \quad (11)$$

In the above equation, $E_c$ represents the semantic encoder, which is similar to the item encoder $E_v$.

*3.3.2 Semantic Metacode Learning.* To obtain a semantic metacode that provides a good initialization for the semantic codebook, our strategy involves using the weights of a semantic encoder, which incorporates substantial homogeneity information for semantic encoding, as the semantic metacode. To acquire the semantic encoder, we need to disentangle the parameter generation into trunk parameters and branch parameters, using semantic sequences and item sequences for generation, respectively. Once training is complete, we can obtain the semantic metacode.

Disentangling the parameter generation model into item-to-parameter and semantic-to-parameter means separating the model parameters into a user model and a user group model. The advantage of the former lies in its extreme personalization, but the downside is the inaccurate parameter generation due to the high heterogeneity of data. The latter has the advantage of low data heterogeneity, allowing for more stable and accurate parameter generation, but its downside is that the personalization level of the semantic sequence is not as high as that of the item sequence. Therefore, how to complement the advantages of both is a question worth considering.

Our solution is to use semantic-to-parameter as the trunk, generating the main part of the model parameters, which can be based on the semantic sequence to obtain the user group model. Then, the parameters generated by item-to-parameter act as branches, limited within a smaller threshold, to ensure the stability of the generated parameters while personalizing from the user group model to the user model. The process can be formulated as follows,

$$\begin{cases} \min_{\Theta_s, \Theta_c, \Theta_v} \mathcal{L}_{\text{SOLID}} = \sum_{u,v,s_c,y \in \mathcal{D}} l_{\text{CE}}(y, \hat{y}), \\ \hat{y} = \mathcal{M}(u, v, s_v; \Theta_s, \Theta_d), \\ \Theta_d = \mathcal{M}_c(E_c(s_c)) + \text{Clip}(\mathcal{M}_v(E_v(s_v)); \mathcal{T}), \end{cases} \quad (12)$$

where $\mathcal{T}$ is a hyperparameter to control the threshold of the generated parameter shift. The training procedure can thus be formulated as the following optimization problem.

*3.3.3 Semantic Codebook Learning.* However, even if we disentangle the model parameters and deal with the item-to-parameter process, the item-to-parameter mapping still needs to be used to generate model parameters. Therefore, to further improve the accuracy of the item-to-parameter mapping, so we designe a Semantic Codebook (SC). Upon obtaining the semantic metacode, we can initialize the semantic codebook with it. Subsequently, we continue using the trunk and branch method of parameter generation, specifically semantic-to-parameter and item-to-parameter, to derive the parameters for the dynamic layer of the model. In the branch branch, the item representations are replaced with semantic codes from the codebook, which are then used to further predict model parameters. The generated model parameters are used for click prediction on item sequences, just as before, ultimately allowing for the training of the semantic codebook. The specific method for computing the loss is described below. SC is denoted as $D$, and $D \in \mathbb{R}^{\mathcal{N}_c \times L_r}$. Specifically, we first use the weights of the semantic encoder in the semantic-to-parameter to initialize the item representation, as their dimensions are the same. Then, we encode the user's item representation. For a piece of data, as introduced in the notation description section, its item representation is $\mathbf{E}_v = \{e_v^1, e_v^2, ..., e_v^{L_s}\}$.

Afterward, we find the closest feature in the SC to replace each item representation in the set $\mathbf{E}_v$, obtaining $\mathbf{E}'_v = \{e'^1_v, e'^2_v, ..., e'^{L_s}_v\}$, and the sequence feature obtained from $\mathbf{E}'_v$ is $e'_v$. Subsequently, we compute the MSE loss between the item representation set $\mathbf{E}'_v$ obtained from the SC and the original set $\mathbf{E}_v$, and incorporate it into the training process as follows,

$$
\begin{cases}
\min_{\Theta_s,\Theta_c,\Theta_v} \mathcal{L}_{\text{SOLID}} = \sum_{u,v,s_c,y \in \mathcal{D}} l_{\text{CE}}(y, \hat{y}) + \lambda l_{\text{MSE}}(\mathbf{E}_v, \mathbf{E}'_v), \\
\qquad\qquad \hat{y} = \mathcal{M}(u, v, s_v; \Theta_s, \Theta_d), \\
\qquad\qquad \Theta_d = \mathcal{M}_c(e_c) + \text{Clip}(\mathcal{M}_v(e'_v)); \mathcal{T}),
\end{cases}
\tag{13}
$$

where $l_{\text{MSE}}$ represents the MSE loss function, and the $\lambda$ is a hyperparameter.

## 4 EXPERIMENTS

### 4.1 Experimental Setup

*4.1.1 Datasets and Preprocessing.* We evaluate SOLID and baselines on eight datasets: Amazon Arts (Arts), Amazon Instruments (Instruments), Amazon Office (Office), Amazon Scientific (Scientific), Amazon CDs (CDs), Amazon Electronic (Electronic), Douban Book (Book), and Douban Music (Music). Arts, Instruments, Office, and Scientific are four benchmarks that was recently released but has been widely used in the multimodal recommendation tasks [19]. CDs, Electronic, Book, and Music are four widely used public benchmarks in the recommendation tasks. The details of these datasets and preprocessing methods can be found in the Appendix. We choose the leave-one-out approach to process the dataset, taking the last action of each user as the test and all previous actions as the train. Our task is CTR (Click-through Rate) prediction, so we process these datasets into CTR prediction datasets. These datasets consist of user rating datasets with complete reviews. We treat all user-item interactions in the dataset as positive samples because having a rating implies that the user clicked on the item. Further, to ensure the training process goes smoothly with both positive and negative samples, we sample 4 negative samples for each positive sample in the training set and 99 negative samples for each positive sample in the test set.

*4.1.2 Baselines.* To evaluate the effectiveness of our method, we selected baselines from the following multiple categories:

- **Static Recommendation Models.**
  **DIN** [29], **GRU4Rec** [9], **SASRec** [11], and **BERT4Rec** [17] are all highly prevalent sequential recommendation methods in both academic research and the industry. They each incorporate different techniques, such as Attention, GRU (Gated Recurrent Unit), and Self-Attention, to enhance the recommendation process.

- **Dynamic Recommendation Models.**
  **DUET** [13] and **APG** [26] consists of two parts: a parameter generation model and a main model. The main model refers to the aforementioned models like DIN, GRU4Rec, SASRec, BERT4Rec, etc. After pre-training, the parameter generation model can generate model parameters for the main model during inference based on the sample.

*4.1.3 Evaluation Metrics.* In the experiments, we use the widely adopted **AUC**, **UAUC**, **NDCG**, and **Recall** as the metrics to evaluate model performance. They are defined as follows,

$$
\text{AUC} = \sum_{x^+ \in \mathcal{D}^+_{\text{Test}}} \sum_{x^- \in \mathcal{D}^-_{\text{Test}}} \frac{\mathbb{1}[\mathcal{M}(x^-) < \mathcal{M}(x^+)]}{|\mathcal{D}^+_{\text{Test}}||\mathcal{D}^-_{\text{Test}}|},
\tag{14}
$$

$$
\text{UAUC} = \frac{1}{|\mathcal{U}|} \sum_{u \in \mathcal{U}} \frac{\sum_{x^{u,+} \in \mathcal{D}^{u,+}_{\text{Test}}} \sum_{x^{u,-} \in \mathcal{D}^{u,-}_{\text{Test}}} \mathbb{1}[\mathcal{M}(x^{u,-}) < \mathcal{M}(x^{u,+})]}{|\mathcal{D}^+_{\text{Test}}||\mathcal{D}^-_{\text{Test}}|},
\tag{15}
$$

$$
\text{NDCG}@K = \frac{1}{|\mathcal{U}|} \sum_{(u,v,s,y) \in \mathcal{D}_{\text{Test}}} \frac{2^{\mathbb{1}(\text{sort}_K(\hat{y}))} - 1}{\log_2(\mathbb{1}(\text{sort}_K(\hat{y})) + 1)},
\tag{16}
$$

$$
\text{Recall}@K = \frac{\sum_{(u,v,s,y) \in \mathcal{D}_{\text{Test}}} \mathbb{1}(\text{sort}_K(\hat{y}))}{|\mathcal{U}|},
\tag{17}
$$

In the equation above, $\mathcal{M}$ represents the model, $\mathbb{1}(\cdot)$ is the indicator function. $\mathcal{D}^+_{\text{Test}}$ and $\mathcal{D}^-_{\text{Test}}$ represent the positive testing dataset and negative testing dataset respectively. $\mathcal{D}^{u,+}_{\text{Test}}$ and $\mathcal{D}^{u,-}_{\text{Test}}$ represent the each user's positive testing dataset and negative testing dataset respectively, $x$, $x^+$, $x^-$, $x^{u,+}$, $x^{u,-}$ represent samples, positive samples, negative samples, each user's positive samples and negative samples, respectively. $\text{sort}_K$ is an operator that sorts items by their scores and retrieves the top-$K$ items with the highest scores. $\mathcal{U}$ is the user set.

### 4.2 Overall Results

As shown in Table 1, we evaluate the overall performance across four multimodal datasets: Arts, Instruments, Office, and Scientific. For each dataset, we test the performance of four SR Models: DIN, GRU4Rec, SASRec, and BERT4Rec. We measure performance using six metrics: AUC, UAUC, NDCG@10, Recall@10, NDCG@20, and Recall@20. For each SR Model, there are five options for DSR Models: none ("-"), APG, Ours (APG), DUET, and Ours (DUET), where "-" indicates no DSR Model usage, i.e., the inherent performance of the SR Model itself. Since the "-" option consistently performs worse than using a DSR Model, our comparison primarily focuses on the performance of APG vs. Ours (APG) and DUET vs. Ours (DUET) for each SR Model. Across all datasets, all SR Models, and all metrics, our proposed methods significantly outperform both APG and DUET.

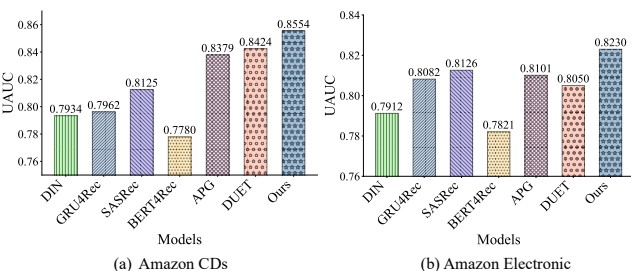

(a) Amazon CDs      (b) Amazon Electronic

**Figure 3: UAUC comparison of the proposed method and baseline on the CDs and Electronic datasets.**

As shown in Figure 3 and Figure 4, we conducted experiments on four other commonly used recommendation datasets and compared the UAUC metric in the figures. Optimal values were used for all

**Table 1: Performance of all selected baseline results including SR Models and DSR methods under P5. The best results is in bold. T-test shows that $p$-value $< 0.05$.**

| SR Model | DSR Model | Arts AUC | UAUC | NDCG@10 | Recall@10 | NDCG@20 | Recall@20 | SR Model | DSR Model | Instruments AUC | UAUC | NDCG@10 | Recall@10 | NDCG@20 | Recall@20 |
|---|---|---|---|---|---|---|---|---|---|---|---|---|---|---|---|
| DIN | - | 0.8193 | 0.7559 | 0.2646 | 0.4696 | 0.2993 | 0.6054 | DIN | - | 0.7974 | 0.7463 | 0.2620 | 0.4576 | 0.2966 | 0.5991 |
| | APG | 0.8432 | 0.7786 | 0.2868 | 0.5024 | 0.3221 | 0.6363 | | APG | 0.8183 | 0.7534 | 0.2680 | 0.4606 | 0.3025 | 0.5962 |
| | Ours (APG) | **0.8459** | **0.7873** | **0.2907** | **0.5144** | **0.3271** | **0.6529** | | Ours (APG) | **0.8274** | **0.7769** | **0.2918** | **0.5006** | **0.3257** | **0.6364** |
| | DUET | 0.8338 | 0.7647 | 0.2837 | 0.4893 | 0.3185 | 0.6202 | | DUET | 0.8126 | 0.7499 | 0.2727 | 0.4658 | 0.3060 | 0.5970 |
| | Ours (DUET) | 0.8426 | 0.7830 | 0.3014 | 0.5162 | 0.3363 | 0.6486 | | Ours (DUET) | 0.8207 | 0.7613 | 0.2850 | 0.4885 | 0.3183 | 0.6181 |
| GRU4Rec | - | 0.8434 | 0.7837 | 0.2799 | 0.4943 | 0.3169 | 0.6380 | GRU4Rec | - | 0.8103 | 0.7604 | 0.2770 | 0.4772 | 0.3102 | 0.6105 |
| | APG | 0.8416 | 0.7796 | 0.2828 | 0.4986 | 0.3196 | 0.6403 | | APG | 0.8171 | 0.7578 | 0.2746 | 0.4716 | 0.3089 | 0.6069 |
| | Ours (APG) | 0.8463 | 0.7897 | 0.3023 | 0.5242 | 0.3378 | 0.6589 | | Ours (APG) | **0.8296** | **0.7752** | **0.2911** | **0.4971** | **0.3265** | **0.6360** |
| | DUET | 0.8463 | 0.7809 | 0.2911 | 0.5061 | 0.3277 | 0.6430 | | DUET | 0.8236 | 0.7568 | 0.2699 | 0.4655 | 0.3058 | 0.6059 |
| | Ours (DUET) | **0.8466** | **0.7915** | **0.3111** | **0.5368** | **0.3460** | **0.6694** | | Ours (DUET) | 0.8261 | 0.7740 | 0.2958 | 0.4987 | 0.3313 | 0.6401 |
| SASRec | - | 0.8383 | 0.7737 | 0.2758 | 0.4852 | 0.3127 | 0.6273 | SASRec | - | 0.8201 | 0.7586 | 0.2729 | 0.4705 | 0.3071 | 0.6051 |
| | APG | 0.8370 | 0.7687 | 0.2816 | 0.4884 | 0.3166 | 0.6222 | | APG | 0.8200 | 0.7523 | 0.2663 | 0.4601 | 0.3010 | 0.5929 |
| | Ours (APG) | 0.8414 | 0.7820 | 0.3018 | 0.5145 | 0.3365 | 0.6468 | | Ours (APG) | 0.8234 | 0.7573 | 0.2699 | 0.4622 | 0.3065 | 0.6029 |
| | DUET | 0.8345 | 0.7660 | 0.2727 | 0.4763 | 0.3101 | 0.6177 | | DUET | 0.8241 | 0.7599 | 0.2768 | 0.4760 | 0.3105 | 0.6076 |
| | Ours (DUET) | **0.8469** | **0.7867** | **0.3022** | **0.5216** | **0.3382** | **0.6560** | | Ours (DUET) | **0.8270** | **0.7661** | **0.2843** | **0.4827** | **0.3198** | **0.6206** |
| BERT4Rec | - | 0.8322 | 0.7791 | 0.2752 | 0.4885 | 0.3126 | 0.6370 | BERT4Rec | - | 0.7951 | 0.7582 | 0.2794 | 0.4723 | 0.3132 | 0.6110 |
| | APG | 0.8485 | 0.7848 | 0.2986 | 0.5123 | 0.3346 | 0.6478 | | APG | 0.8261 | 0.7650 | 0.2895 | 0.4891 | 0.3226 | 0.6202 |
| | Ours (APG) | **0.8504** | **0.7921** | **0.3054** | **0.5279** | **0.3411** | **0.6631** | | Ours (APG) | **0.8386** | **0.7846** | **0.3058** | **0.5179** | **0.3412** | **0.6568** |
| | DUET | 0.8454 | 0.7834 | 0.2861 | 0.5025 | 0.3238 | 0.6424 | | DUET | 0.8285 | 0.7686 | 0.2712 | 0.4750 | 0.3078 | 0.6191 |
| | Ours (DUET) | 0.8497 | 0.7970 | 0.3088 | 0.5344 | 0.3456 | 0.6748 | | Ours (DUET) | 0.8326 | 0.7811 | 0.2992 | 0.5104 | 0.3329 | 0.6435 |

| SR Model | DSR Model | Office AUC | UAUC | NDCG@10 | Recall@10 | NDCG@20 | Recall@20 | SR Model | DSR Model | Scientific AUC | UAUC | NDCG@10 | Recall@10 | NDCG@20 | Recall@20 |
|---|---|---|---|---|---|---|---|---|---|---|---|---|---|---|---|
| DIN | - | 0.8158 | 0.7510 | 0.2701 | 0.4702 | 0.3046 | 0.6045 | DIN | - | 0.6100 | 0.5971 | 0.1337 | 0.2609 | 0.1648 | 0.3880 |
| | APG | 0.8359 | 0.7639 | 0.2862 | 0.4903 | 0.3202 | 0.6202 | | APG | 0.7310 | 0.6969 | 0.1700 | 0.3238 | 0.2099 | 0.4816 |
| | Ours (APG) | **0.8394** | **0.7673** | 0.2764 | 0.4823 | 0.3128 | 0.6222 | | Ours (APG) | **0.7315** | **0.6989** | **0.1746** | **0.3429** | **0.2147** | **0.5020** |
| | DUET | 0.8297 | 0.7531 | 0.2813 | 0.4816 | 0.3147 | 0.6085 | | DUET | 0.6714 | 0.6266 | 0.1428 | 0.2736 | 0.1748 | 0.3979 |
| | Ours (DUET) | 0.8361 | 0.7642 | **0.2949** | **0.4970** | **0.3282** | **0.6240** | | Ours (DUET) | 0.7138 | 0.6682 | 0.1589 | 0.3012 | 0.1989 | 0.4573 |
| GRU4Rec | - | 0.8346 | 0.7606 | 0.2704 | 0.4762 | 0.3055 | 0.6117 | GRU4Rec | - | 0.7424 | 0.7094 | 0.1621 | 0.3214 | 0.2049 | 0.4952 |
| | APG | 0.8343 | 0.7623 | 0.2809 | 0.4831 | 0.3154 | 0.6159 | | APG | 0.7273 | 0.6933 | 0.1592 | 0.3159 | 0.1988 | 0.4758 |
| | Ours (APG) | 0.8354 | 0.7671 | 0.2914 | 0.4966 | 0.3255 | 0.6272 | | Ours (APG) | **0.7402** | **0.7133** | **0.1859** | **0.3535** | **0.2273** | **0.5161** |
| | DUET | 0.8399 | 0.7649 | 0.2930 | 0.4976 | 0.3268 | 0.6262 | | DUET | 0.7270 | 0.6881 | 0.1658 | 0.3224 | 0.2036 | 0.4703 |
| | Ours (DUET) | **0.8437** | **0.7737** | **0.3072** | **0.5112** | **0.3403** | **0.6366** | | Ours (DUET) | 0.7410 | 0.7054 | 0.1792 | 0.3415 | 0.2196 | 0.5020 |
| SASRec | - | 0.8288 | 0.7587 | 0.2820 | 0.4858 | 0.3153 | 0.6151 | SASRec | - | 0.7175 | 0.6772 | 0.1587 | 0.3145 | 0.1960 | 0.4631 |
| | APG | 0.8377 | 0.7603 | 0.2823 | 0.4804 | 0.3170 | 0.6117 | | APG | 0.6952 | 0.6610 | 0.1523 | 0.3040 | 0.1910 | 0.4583 |
| | Ours (APG) | 0.8402 | 0.7679 | **0.2997** | 0.4995 | 0.3333 | 0.6269 | | Ours (APG) | **0.7161** | 0.6728 | **0.1634** | **0.3122** | **0.2002** | 0.4580 |
| | DUET | 0.8395 | 0.7594 | 0.2833 | 0.4831 | 0.3173 | 0.6105 | | DUET | 0.6992 | 0.6565 | 0.1579 | 0.3040 | 0.1944 | 0.4481 |
| | Ours (DUET) | **0.8460** | **0.7735** | **0.2997** | **0.5061** | **0.3345** | **0.6380** | | Ours (DUET) | 0.7111 | **0.6738** | 0.1548 | 0.3016 | 0.1957 | **0.4614** |
| BERT4Rec | - | 0.8184 | 0.7544 | 0.2701 | 0.4732 | 0.3049 | 0.6092 | BERT4Rec | - | 0.7329 | 0.7000 | 0.1744 | 0.3306 | 0.2108 | 0.4768 |
| | APG | 0.8354 | 0.7633 | 0.2885 | 0.4923 | 0.3223 | 0.6222 | | APG | 0.7255 | 0.6953 | 0.1699 | 0.3306 | 0.2069 | 0.4758 |
| | Ours (APG) | **0.8462** | **0.7767** | **0.3032** | **0.5130** | **0.3374** | **0.6419** | | Ours (APG) | **0.7456** | **0.7132** | 0.1760 | **0.3508** | 0.2183 | **0.5167** |
| | DUET | 0.8371 | 0.7682 | 0.2842 | 0.4900 | 0.3187 | 0.6223 | | DUET | 0.7325 | 0.6962 | 0.1707 | 0.3262 | 0.2090 | 0.4785 |
| | Ours (DUET) | 0.8380 | 0.7731 | 0.2892 | 0.4987 | 0.3249 | 0.6365 | | Ours (DUET) | 0.7420 | 0.7108 | **0.1826** | 0.3477 | **0.2235** | 0.5099 |

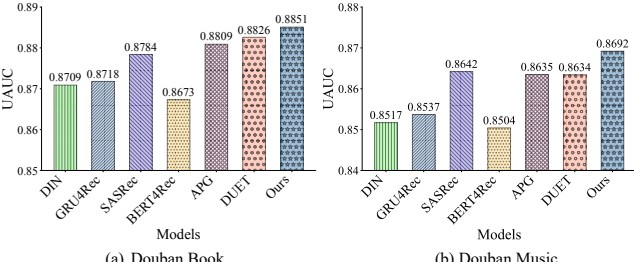

(a) Douban Book     (b) Douban Music

**Figure 4: UAUC comparison of the proposed method and baseline on the `Book` and `Music` datasets.**

SR Models and DSR Models. Our method significantly outperforms other SR and DSR Models across all the datasets.

## 4.3 Ablation Study

We conduct ablation studies on each module and each modality to further analyze the impact of them. Here, we choose the Arts dataset, the SASRec model for SR, and the DUET model for DSR. Each row's ✓ and ✗ respectively indicate with and without the module/modality.

*4.3.1 Ablation Study on Modules.* As shown in Table 2, we conduct an ablation study on each module proposed in our method, SPG stands for Semantic Parameter Generation, SML stands for Semantic Metacode Learning, and SCL stands for Semancic Codebook Learning. Since SPG is a prerequisite for SML, SML cannot exist independently of SPG; therefore, there is no separate performance data for SML alone in the table. The translation from Chinese to English for your text is: The first line represents the traditional DSR model where parameters are generated using an item sequence. The second line represents generating parameters using a semantic sequence. The third line represents the joint generation of parameters using both item sequence and semantic sequence, with joint training. The fourth line represents using semantic codebook learning without using semantic information. The fifth line represents

our complete method. The experiments show that the model performs best when all three modules are used. In terms of individual modules, SCL has the greatest impact on performance.

**Table 2: Results of the ablation study over our proposed methods with respect to the modules. The best results is in bold.**

| Module | | | Metrics | | | | | |
|---|---|---|---|---|---|---|---|---|
| SPG | SML | SCL | AUC | UAUC | NDCG@10 | Recall@10 | NDCG@20 | Recall@20 |
| ✗ | ✗ | ✗ | 0.8345 | 0.7660 | 0.2727 | 0.4763 | 0.3101 | 0.6177 |
| ✓ | ✗ | ✗ | 0.8459 | 0.7783 | 0.2905 | 0.5069 | 0.3270 | 0.6425 |
| ✓ | ✓ | ✗ | 0.8270 | 0.7530 | 0.2491 | 0.4539 | 0.2857 | 0.5922 |
| ✗ | ✗ | ✓ | 0.8461 | 0.7828 | 0.2976 | 0.5166 | 0.3326 | 0.6481 |
| ✓ | ✓ | ✓ | **0.8469** | **0.7867** | **0.3022** | **0.5216** | **0.3382** | **0.6560** |

*4.3.2 Ablation Study on Modalities.* As shown in Table 3, we conduct ablation study on each modality. The experimental results show that the fusion of three modalities—ID, Image, and Text—is not necessarily the best option. In terms of the impact on performance for individual modalities, Text > Image > ID. For the fusion of two modalities, in terms of impact on performance, ID + Text > Image + Text > ID + Image.

**Table 3: Results of the ablation study over our proposed methods with respect to the modalities. The best results is in bold.**

| Modality | | | Metrics | | | | | |
|---|---|---|---|---|---|---|---|---|
| ID | Image | Text | AUC | UAUC | NDCG@10 | Recall@10 | NDCG@20 | Recall@20 |
| ✓ | ✗ | ✗ | 0.8419 | 0.7770 | 0.2527 | 0.3814 | 0.2884 | 0.5024 |
| ✗ | ✓ | ✗ | 0.8450 | 0.7824 | 0.2579 | 0.3867 | 0.2947 | 0.5106 |
| ✗ | ✗ | ✓ | **0.8480** | 0.7861 | 0.2611 | 0.3929 | 0.2977 | 0.5156 |
| ✓ | ✓ | ✗ | 0.8402 | 0.7728 | 0.2507 | 0.3781 | 0.2872 | 0.5004 |
| ✓ | ✗ | ✓ | 0.8478 | **0.7863** | **0.2631** | **0.3959** | **0.3000** | **0.5201** |
| ✗ | ✓ | ✓ | 0.8449 | 0.7816 | 0.2574 | 0.3879 | 0.2929 | 0.5082 |
| ✓ | ✓ | ✓ | 0.8461 | 0.7828 | 0.2603 | 0.3904 | 0.2979 | 0.5166 |

## 4.4 Hyperparameter Analysis

We conduct experimental analysis on the main hyperparameters $\lambda$ and $\mathcal{T}$, as well as their grid search.

*4.4.1 The Impact of $\lambda$.* As shown in Table 4, we fix $\mathcal{T} = 0.01$ and vary $\lambda$ to observe changes in performance, comparing it across six metrics. $\lambda$ is a hyperparameter that influences the fusion weight between the MSE (Mean Squared Error) loss in recommendation task learning and the CE (Cross Entropy) loss in semantic codebook learning. The results indicate that the performance is optimal when $\lambda = 0.1$, suggesting that at this setting, the model achieves the best joint learning effect for the recommendation task and the semantic codebook.

**Table 4: The impact of the hyperparameter $\lambda$ on performance. The best results is in bold.**

| $\lambda$ | Metrics | | | | | |
|---|---|---|---|---|---|---|
| | AUC | UAUC | NDCG@10 | Recall@10 | NDCG@20 | Recall@20 |
| 1 | 0.8485 | 0.7839 | 0.2901 | 0.5103 | 0.3273 | 0.6466 |
| 0.1 | **0.8479** | **0.7850** | **0.2983** | **0.5155** | **0.3347** | **0.6510** |
| 0.01 | 0.8489 | 0.7829 | 0.2952 | 0.5120 | 0.3317 | 0.6459 |
| 0.01 | 0.8468 | 0.7810 | 0.2937 | 0.5103 | 0.3292 | 0.6431 |

*4.4.2 The Impact of $\mathcal{T}$.* As shown in Table 5, we fix $\lambda = 0.1$ and vary $\mathcal{T}$ to observe changes in performance, conducting comparisons across six metrics. $\mathcal{T}$ is a hyperparameter that affects the fusion weight between homogeneous information, i.e., semantic sequence to parameter, and heterogeneous information, i.e., item sequence to parameter. The results show that the optimal performance is achieved when $\mathcal{T} = 0.01$, indicating that at this point, the fusion effect between semantic sequence to parameter and item sequence to parameter is the best.

**Table 5: The impact of the hyperparameter $\mathcal{T}$ on performance. The best results is in bold.**

| $\mathcal{T}$ | Metrics | | | | | |
|---|---|---|---|---|---|---|
| | AUC | UAUC | NDCG@10 | Recall@10 | NDCG@20 | Recall@20 |
| 1 | 0.7785 | 0.7308 | 0.2220 | 0.4114 | 0.2574 | 0.5576 |
| 0.1 | 0.8460 | 0.7822 | 0.2907 | 0.5088 | 0.3273 | 0.6446 |
| 0.01 | **0.8479** | **0.7850** | **0.2983** | **0.5155** | **0.3347** | **0.6510** |
| 0.001 | 0.8451 | 0.7801 | 0.2925 | 0.5078 | 0.3279 | 0.6402 |

*4.4.3 Hyperparameter Grid Search.* As shown in Figure 5, the horizontal axis represents $\lambda$, and the vertical axis represents $\mathcal{T}$. The depth of the color and the radius of the circle represent the magnitude of the value; the larger the value, the deeper the color and the larger the circle (i.e., the larger the radius). Blue, green, and orange represent the metrics UAUC, NDCG@10, and Recall@10, respectively. The results show that the best performance is achieved when $\lambda = 0.1$ and $\mathcal{T} = 0.01$.

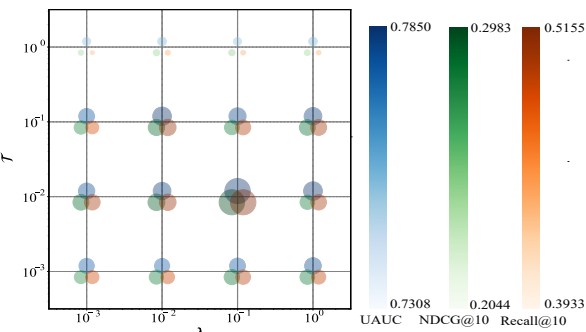

**Figure 5: Hyperparameter Grid Search.**

## 5 CONCLUSION

In this paper, we have presented the Semantic Codebook Learning for Dynamic Recommendation Models (SOLID) as a solution to the limitations faced by existing dynamic sequence recommendation systems (DSR). Our framework integrates multimodal information, including images and text, with user-item interactions to enhance recommendation accuracy and adaptability. By disentangling model parameters into trunk parameters capturing generalized user behavior trends and branch parameters tailored to individual user actions, SOLID offers a more efficient and effective recommendation system. Through extensive experimentation across multiple datasets, we have demonstrated that SOLID significantly outperforms previous DSR models, with an significant improvement on extensive datasets and models. These results underscore the potential of leveraging multimodal information to advance the capabilities of dynamic recommendation systems, paving the way for more personalized and responsive user experiences in the era of digital personalization.

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
