# OpenReview forum: "Semantic Codebook Learning for Dynamic Recommendation Models"
_acmmm.org/ACMMM/2024/Conference — MM2024 Poster_

### Official Review · Reviewer_WDiW · 2024-05-10

**Rating:** 2
**Confidence:** 2

**Summary:**

This paper proposes a method named SOLID to address issues in conventional dynamic sequential recommendation models, such as instability due to minor variations in item sequences and difficulties in learning item representations due to the sparsity of user-item interactions. SOLID generates parameters based on the semantic sequences of each item, demonstrating improved accuracy in dynamic recommendation models APG and DUET when enhanced with SOLID.

**Strengths:**

While I am not a specialist in dynamic sequential recommendation models, the idea of transforming item sequences into semantic sequences to solve problems inherent in traditional models seems reasonable. The publication of the source code is also commendable.

**Limitations:**

Due to insufficient experimental details, I found it difficult to accept this paper:
1. As noted in Section 4.4 and the Appendix, the proposed method involves several hyperparameters. However, how these values were determined for the results in Table 1 is not described. Section 4.1.1 only mentions dividing the dataset into training and test data, without any mention of validation data. To prevent readers from suspecting that hyperparameters were tuned using the test data, it is essential to create and use validation data for hyperparameter tuning.
2. In Figures 3 and 4, it is unclear which methods labeled as "Ours" were used for SR and DSR.
3. The ablation study in Section 4.3.2 concludes that using both ID and Text yields the highest accuracy, yet it is unclear which data (ID, Image, Text) were used in Table 1. The values in Table 2 for SPG+SML+SCL match those in Table 1 for Arts in SASRec+Ours(DUET); however, Table 3 lacks a corresponding row. In addition, why are the NDCG and Recall values in Table 3 significantly lower than those in Table 1?
4. The ablation study reports only on the Arts dataset results for SASRec+Ours(DUET). Were similar results obtained for other datasets, and with other SR and DSR? To avoid suspicion that only favorable results were reported, it should be clarified if similar results were observed.
5. Section 4.3 mentions using SASRec+Ours(DUET) on the Arts dataset for evaluation, but Section 4.4 does not specify the datasets, SR, or DSR used. It concludes that the best accuracy was achieved with $\lambda = 0.1$ and $\tau = 0.01$, but it is unclear whether these values were optimal for all datasets and methods.

Minor Comments:
1. In Figure 1(d), it appears as if "Ours" achieved a UAUC and Recall@10 of 1.0 across all datasets, which could be misleading. If all values are normalized based on the value of “Ours”, then the Arts data set for Recall@10 has a DIN value of about 0.1. However, looking at the results in Table 1, the Recall@10 value of DIN is still inaccurate because it is about 90% of the value of the proposed method.
2. The definition of "Clip" in Equations 12 and 13 is not described.
3. The intention behind the phrase "The translation from Chinese to English for your text is:" in Section 4.3.1 is unclear.

**Suitability:**

2

---

### Official Review · Reviewer_qgia · 2024-05-23

**Rating:** 3
**Confidence:** 3

**Summary:**

This article primarily introduces a framework called Semantic Codebook Learning for Dynamic Recommendation Models (SOLID), aimed at addressing challenges faced by Dynamic Sequential Recommendation (DSR) systems. DSR systems, which adjust their parameters in real time to adapt to users' evolving preferences, offer significant advancements over traditional static models.

**Strengths:**

1.The authors identified limitations within the existing Dynamic Sequential Recommendation (DSR) paradigm, indicating a clear understanding of the challenges in the field.
2.The introduction of the SOLID framework, coupled with a semantic codebook, represents an innovative solution to address the deficiencies observed in the DSR paradigm.

**Limitations:**

Q1: How can we demonstrate that the method proposed in this paper, shifting the modeling approach from item-based to semantics-based parameter generation, addresses the complexity and instability issues mentioned in the motivation?It would be beneficial to provide experimental results regarding the amount of parameters saved and the time-space saved compared to the baseline.

Q2: While slight variations in behavior sequences may alter the modeling of item-to-parameter relationships, what is the magnitude of their impact on the overall performance of the model?

Q3:The complexity of the latent space has been reduced, but what is the cost of the semantic codebook? Could you provide a comparison of complexity before and after?

Q4:Could you please check the writing of the last paragraph in the introduction to make it easier to understand and structurally stronger?

Q5:Based on the ablation experiments, it is evident that the performance significantly deteriorates when modal information is not utilized. In comparison to traditional methods, the introduced approach incorporates additional modal information.  these pieces of information could primarily contribute to the performance improvement, rather than the two main reasons stated by the authors, especially considering that the semantic codebook's ability to better learn homogeneity signals, etc., is not extensively discussed in the paper. It is suggested that the authors include an experiment comparing the direct addition of modal features to traditional DSR methods without using a semantic codebook to observe the resulting effects.

**Suitability:**

3

---

### Official Review · Reviewer_96Rr · 2024-05-23

**Rating:** 6
**Confidence:** 4

**Summary:**

This manuscript focuses on the problem of changing users' preferences while generating recommendations. The semantic codebook is used to replace some old parameters with the current encoder parameters.

**Strengths:**

1. Presentation
2. Experiments including the improvements.
3. The topic is interesting.

**Limitations:**

None.

**Suitability:**

3

---

### Meta-Review · Area_Chair_3fRv · 2024-07-02

**Recommendation:** Accept (Poster)
**Confidence:** 4

**Metareview:**

Reviewer 96Rr's feedback is too brief, and despite requests for further expansion, the reviewer declined to elaborate, leading to the exclusion of the comments from the decision-making process. Besides, because reviewers qgia and WDiW offer divergent perspectives, I have carefully examined the manuscript and the authors' rebuttal. While the manuscript's motivation is not sufficiently robust, the overall methodology is novel, and the experiments are well-executed. The authors' detailed responses to the critiques of reviewers qgia and WDiW are particularly noteworthy. Based on these considerations, I recommend the acceptance of this paper. It is advised, however, that the authors carefully address the further inquiries posed by Reviewer WDiW in their subsequent revisions.